# Accessing the bottleneck in all-solid state batteries, lithium-ion transport over the solid-electrolyte-electrode interface

Chuang Yu[1], Swapna Ganapathy[1], Ernst R.H. Van Eck[2], Heng Wang[1], Shibabrata Basak[3], Zhaolong Li[1] & Marnix Wagemaker[1]

Solid-state batteries potentially offer increased lithium-ion battery energy density and safety as required for large-scale production of electrical vehicles. One of the key challenges toward high-performance solid-state batteries is the large impedance posed by the electrode–electrolyte interface. However, direct assessment of the lithium-ion transport across realistic electrode–electrolyte interfaces is tedious. Here we report two-dimensional lithium-ion exchange NMR accessing the spontaneous lithium-ion transport, providing insight on the influence of electrode preparation and battery cycling on the lithium-ion transport over the interface between an argyrodite solid-electrolyte and a sulfide electrode. Interfacial conductivity is shown to depend strongly on the preparation method and demonstrated to drop dramatically after a few electrochemical (dis)charge cycles due to both losses in interfacial contact and increased diffusional barriers. The reported exchange NMR facilitates non-invasive and selective measurement of lithium-ion interfacial transport, providing insight that can guide the electrolyte–electrode interface design for future all-solid-state batteries.

[1] Department of Radiation Science and Technology, Delft University of Technology, Mekelweg 15, 2629 JB Delft, The Netherlands. [2] Institute for Molecules and Materials, Radboud University, Heyendaalseweg 135, 6525 AJ Nijmegen, The Netherlands. [3] Department of Quantum Nanoscience, Kavli Institute of Nanoscience Delft, Delft University of Technology, Lorentzweg 1, 2628 CJ Delft, The Netherlands. Chuang Yu and Swapna Ganapathy contributed equally to this work. Correspondence and requests for materials should be addressed to M.W. (email: m.wagemaker@tudelft.nl)

The high-energy density and long cycle life of lithium-ion batteries has enabled the development of mobile electronic equipment, and recently of electrical vehicles (EV's) and static energy storage to stabilize the grid and balance renewable energy supply and demand. However, the use of liquid organic electrolytes in lithium-ion batteries raises safety issues, in particular for relatively large systems as employed in electrical cars and grid storage. The origin of the safety risk is the gas production and leakage of the flammable liquid organic electrolytes when operating at high voltages and/or elevated temperatures. A potential solution is the use of solid-state electrolytes, a goal which has been pursued for many decades[1–11]. Recently, solid-state lithium-ion battery research has intensified dramatically[4–9, 12–15], propelled by the development of several structural families of highly conductive solid electrolytes, including LISICON like compounds[14, 16–18], argyrodites[19], garnets[20, 21], and NASICON-type structures[22].

In addition to improved battery safety, solid electrolytes potentially offer additional advantages. These include freedom in design of the battery geometry and improvement of the packing efficiency of the cells, which will enable increased practical battery energy densities. Additionally, a number of solid-state electrolytes may offer a larger electrochemical stability window compared to liquid electrolytes or lead to a narrow stable interfacial passivation layer, which facilitates a long cycle life and offers the possibility of employing high-voltage cathodes, which in turn further increases the battery energy density. On the anode side, solid-state batteries open the door to safe application of Li-metal by suppressing dendrite formation, also increasing the energy density.

Despite the great progress in synthesizing excellent lithium-ion conducting solid electrolytes, the rate capability of almost all-solid-state cells is poor, in particular those employing cathodes undergoing a high volume change such as sulfide-based electrodes[23–31] and those utilizing high-voltage cathodes[32, 33]. Despite providing a high bulk lithium-ion conductivity, the poor rate and cycle performance of solid-state batteries are ascribed to a high internal resistance for lithium-ion transfer over the solid–solid electrode–electrolyte interfaces[4–9, 33–39]. Although difficult to ascertain experimentally, the origin of the interfacial resistance will depend on the electrode–electrolyte combination and its preparation route. Both chemical incompatibility and a narrow solid electrolyte electrochemical window[4, 11, 40] may result in an interface layer that poses high resistance toward lithium-ion transport[33–42]. Driven by the potential difference between the positive electrode and electrolyte, the interfaces will induce space charges, potentially leading to local lithium-ion depletion of the electrolyte. This poses an additional hurdle for lithium-ion transport over the solid–solid electrode–electrolyte interface[36, 43]. Finally, perhaps one of the biggest challenges is the mechanical stability, where the volume changes of electrode materials during (dis)charge may cause loss of contact between the electrode and the electrolyte particles, blocking lithium-ion transport across the interface. These challenges indicate that whether or not solid-state batteries will be able to deliver the performance necessary for EVs will depend on the development of stable interfaces that allow facile ionic charge transfer. Several strategies have been developed to improve the interface resistances; an example of which includes coating the electrodes with an oxide barrier layer enabling high-rate cycling[38, 43]. To guide the interfacial design, it is paramount to investigate interface reactions and charge transport over the solid–solid electrode–electrolyte interfaces. The charge transfer resistance is most often estimated by impedance spectroscopy, which appears accurate in well-defined thin film solid-state batteries, but difficult if not impossible in the complex morphologies of bulk solid-state batteries[44]. Using impedance spectroscopy, it is not trivial to distinguish the interface from the bulk lithium-ion conductivity as it probes the charge kinetics over tens of nanometers, including the influence of porosity, grain boundaries, and effects introduced by the contact of the solid electrolyte under investigation with the electrodes.

Nuclear magnetic resonance (NMR) spectroscopy, a non-destructive contactless probe, has been shown to offer unique complementary information to impedance spectroscopy, by its high sensitivity toward the lithium-ion mobility in bulk battery materials[45–48]. An additional opportunity provided by solid-state NMR in multi-phase battery materials, either consisting of multiple electrode phases or a mixture of electrode and electrolyte phases, is the possibility to measure the spontaneous lithium-ion exchange between different lithium-containing phases. This provides unique selectivity for charge transfer over phase boundaries[24, 47, 49], as recently shown to be feasible for the $Li_6PS_5Cl–Li_2S$ solid electrolyte–electrode combination[24].

Here we employ two-dimensional exchange NMR spectroscopy (2D-EXSY) providing unique quantitative insight in the spontaneous exchange between a solid electrolyte and an electrode. Enabled by the difference in NMR chemical shift, the lithium-ion transport was determined over the interface of the $Li_6PS_5Br–Li_2S$ cathode mixture at different stages in the preparation and before and after cycling, giving unprecedented insight into the evolution of the resistance between the solid electrolyte and cathode. Nanosizing $Li_2S$ and establishing intimate contact with the argyrodite $Li_6PS_5Br$ electrolyte is shown to be necessary to provide measurable charge transfer over the interfaces. Although charge transport over the $Li_6PS_5Br–Li_2S$ interfaces is facile, the small amount of contact area in pristine, uncycled, cathodic mixtures, results in an interfacial conductivity that is orders of magnitude smaller than the bulk conductivity. After cycling, the lithium-ion kinetics over the interface dramatically decrease, most likely due to both the large volumetric changes that compromise the interfacial contact and by increased barriers for diffusion due to the formation of side products. Both these factors are responsible for the decrease in capacity during repeated cycling. These observations demonstrate the crucial importance of developing strategies that preserve the interfacial integrity during cycling, and introduce the unique ability of exchange NMR to investigate the interfacial charge transport allowing direct and non-invasive quantification.

## Results

**Impedance spectroscopy and solid-state battery performance.** The argyrodite $Li_6PS_5Br$ solid electrolyte material was prepared by ball milling at 600 rpm for different milling times followed by annealing at 300 °C for 5 h. Impedance spectroscopy of the annealed $Li_6PS_5Br$ shows a room temperature conductivity of 0.011(1) S cm$^{-1}$, comparable to literature values[50]. Relaxation NMR (see Supplementary Fig. 9), which probes the lithium-ion hopping through the bulk lattice, resulted in a conductivity of 0.013(1) S cm$^{-1}$ at 78 °C with an activation energy of 0.10(5) eV, indicating mobility comparable with recently reported NMR results[51], and a larger bulk conductivity than that resulting from impedance spectroscopy. This may indicate that, like for the analogous $Li_6PS_5Cl$, grain boundaries may be responsible for the lower bulk conductivity measured by impedance spectroscopy[24].

To gain insight into how the preparation of the cathode mixtures of the $Li_2S$ cathode and $Li_6PS_5Br$ solid electrolyte affect the lithium-ion transport over the electrode–electrolyte interface, the preparation steps, shown in Fig. 1a, were investigated by both electrochemical impedance spectroscopy (EIS) and $^7Li$ exchange NMR. In addition, the capacity retention was determined by galvanostatic charging where all the mixtures have a 1:1 mass ratio of the $Li_6PS_5Br$ solid electrolyte to the $Li_2S$ cathode material.

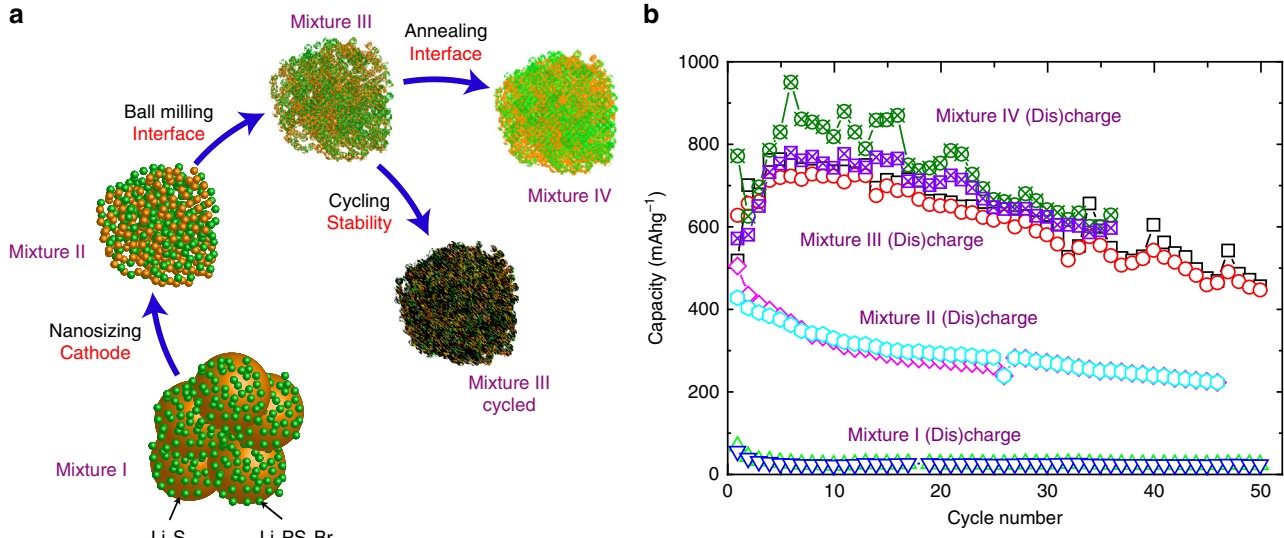

**Fig. 1** Schematic representation of the different stages in solid-state battery cathode preparation and the solid-state battery capacity retention. **a** The different stages in cathode preparation and cycling for which the lithium-ion transport over the $Li_2S$–$Li_6PS_5Br$ interfaces, the charge transfer reaction is measured by $^7Li$ exchange NMR and electrochemical impedance spectroscopy (EIS). **b** The charge/discharge capacity upon cycling of the solid-state Li–S cell using micro-sized $Li_2S$ (mixture I in **a**), nano-$Li_2S$ (mixture II in **a**), mixed nano-$Li_2S$ (mixture III in **a**), and annealed mixed nano-$Li_2S$ (mixture IV in **a**) as the active material. Charge and discharge curves are shown in Supplementary Fig. 3. The charge/discharge current density was set at 0.064 mA cm$^{-2}$ ($5.03 \times 10^{-5}$ A), and the lower and upper voltage cutoff were set to 0 and 3.5 V vs. In

Recent research has shown that both nanosizing $Li_2S$ and intimate mixing with $Li_6PS_5Br$ (mixture III in Fig. 1a) is crucial to obtain high capacities over multiple cycles[27]. This is confirmed in Fig. 1b by comparing the capacity retention of cathode mixture I (120 nm-sized $Li_2S$) with cathode mixture III (38 nm-sized $Li_2S$ according to XRD, see Supplementary Fig. 1) cycled vs. an In foil anode at a current density of 0.064 mA cm$^{-2}$ in the voltage window of 0–3.5 V vs. In (0.62–4.12 V vs. Li$^+$/Li). The crystallite sizes are determined from XRD line broadening shown in Supplementary Fig. 1, which are consistent with TEM observations in Supplementary Fig. 2. The comparison of the first 4 charge/discharge voltage curves in Supplementary Fig. 3 shows that for the large $Li_2S$ particle size in mixture I, no obvious charge plateau is observed, whereas the nano-$Li_2S$ (mixture III) delivers two distinct charge plateaus, located at 1.8 and 2.5 V vs. In, respectively. On discharge, both cathode mixtures show a discharge plateau at 1.4 V vs. In and the nanosize $Li_2S$ mixture displays an additional, albeit ill-defined, plateau between 0.5 and 1.0 V vs. In. Interestingly, nanosized $Li_2S$ shows a slight increase in discharge capacity with increasing cycle number during the first 5 cycles, while the charge and discharge capacity of commercial $Li_2S$ dramatically decrease with increasing cycle numbers, as shown in Fig. 1b, also observed by Nagao et al.[52]. The increase in discharge capacity of the nano-$Li_2S$ cathode is attributed to the activation process of $Li_2S$ occurring during the first few charge/discharge cycles of the solid-state cell. The difference in (dis)charge capacity upon cycling shown in Fig. 1b is striking. Nanosize $Li_2S$ delivers 628 vs. 56 mAh g$^{-1}$ for large-sized $Li_2S$.

The present mixture III provides a higher discharge capacity and better cyclability compared to most reported comparable solid-state cells employing composite $Li_2S$ electrodes in combination with argyrodite and $80Li_2S$–$20P_2S_5$ solid electrolytes[27, 52–54]. However, capacity retention in this work is not as good as that for the solid-state cell in combination of a mixed-conductive $Li_2S$ nanocomposite cathode and $Li_6PS_5Cl$ electrolyte most likely due to the better distribution of $Li_2S$, $Li_6PS_5Cl$, and carbon in the cathode mixture[55].

**Exchange NMR, lithium-ion transport between $Li_2S$ and $Li_6PS_5Br$.** The large impact of the $Li_2S$ particle size, i.e., 120 nm for mixture I and 38 nm for mixture III on the (dis)charge capacity, can be attributed to the poor ionic and electronic conductivity of $Li_2S$ in the first place. Smaller $Li_2S$ particle sizes will reduce charge transport distances reducing the kinetic restrictions of $Li_2S$. Additionally, thorough mixing of the nano-sized $Li_2S$ with the $Li_6PS_5Br$ solid electrolyte will lead to more $Li_2S$–$Li_6PS_5Br$ interfaces. However, to what extent these interfaces allow facile lithium-ion transport depending on the preparation and cycling conditions is too difficult to assess. Electrochemical impedance spectroscopy (EIS) is employed for all mixtures I–IV before and after cycling, as shown in Supplementary Fig. 4. Assuming simple equivalent circuits, $R_1(R_2Q_2)Q^{56}$, $R_1(R_2Q_2)(R_3Q_3)Q_4$, and $R_1(R_2Q_2)(Q_3(R_3W))$, the EIS data were fitted, in each case not resulting in very good fits, see Supplementary Fig. 4 and Supplementary Table 1. Given the small number of parameters, and aiming at accurate determination of $R_1$, the bulk solid electrolyte resistance, and $R_2$, the grain boundary resistance between the solid electrolyte and the $Li_2S$ cathode, the best choice appears to be the most simple circuit, $R_1(R_2Q_2)Q^{56}$ for the pristine mixtures. For the cycled mixtures, an additional semicircle indicates the establishment of an another interface, possible at the In anode, which was most accurately fit by the $R_1(R_2Q_2)(R_3Q_3)Q_4$ equivalent circuit. The resulting values for the $R_1$ and $R_2$ resistances are reported in Supplementary Table 1.

The results $R_1$ and $R_2$ from EIS in Supplementary Table 1 show that, for all mixtures, cycling leads to an increase of both the bulk and interface impedance. The increase in interface impedance is most likely the consequence of a combination of electrochemical reactions at the $Li_6PS_5Br$–$Li_2S$ interface and loss of interfacial contact due to the volumetric changes of the $Li_2S$ cathode upon cycling as will be discussed in more detail below. The results in Supplementary Table 1, indicate that nanosizing, going from mixture I to II, and mixing, going from mixture II to III, has only minor influences on the interfacial resistance. This is difficult to explain because the amount of $Li_6PS_5Br$–$Li_2S$ interface area is expected to change significantly upon nanosizing and mixing.

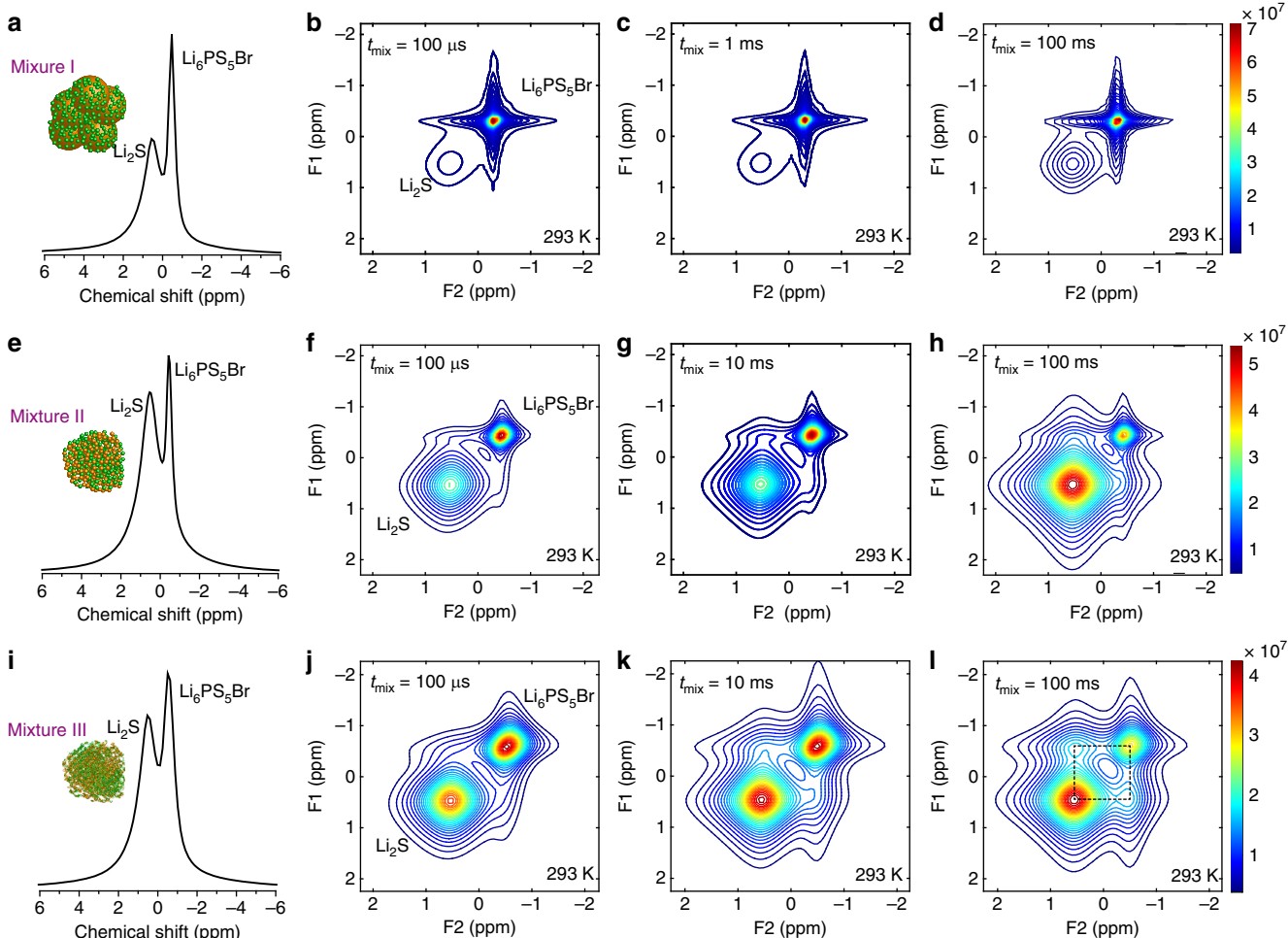

**Fig. 2** NMR measuring the spontaneous lithium-ion transport between the $Li_6PS_5Br$ solid electrolyte and the $Li_2S$ cathode. **a**, **e**, **i** One-dimensional (1D) [7]Li magic angle spinning (MAS) spectra corresponding to the $Li_6PS_5Br$–$Li_2S$ cathode mixtures **a** mixture I, where $Li_2S$ is micron-sized **e** mixture II, where $Li_2S$ is nanosized **i** mixture III, where nanosized $Li_2S$ is thoroughly mixed with $Li_6PS_5Br$. **b-d**, **f-h**, **j-l** Two-dimensional (2D) [7]Li-[7]Li exchange spectra (2D-EXSY) recorded at a [7]Li resonance frequency of 155.506 MHz and a spinning speed of 20 kHz at room temperature for short (100 μs) to long mixing times $t_{mix}$ (100 ms) for **b**, **c**, **d** mixture I, **f**, **g**, **h** for mixture II, and **j**, **k**, **l** for mixture III. For both mixtures I and II, no obvious off-diagonal cross-peak intensity is observed, indicating that the exchange over the solid–solid $Li_6PS_5Br$–$Li_2S$ is very small (based on whether an upper limit for the conductivity can be determined as shown in Fig. 4). For mixture III, the off-diagonal cross-peaks appear at $t_{mix} = 10$ ms, and are most pronounced at $t_{mix} = 100$ ms, and they correspond to lithium-ion exchange from $Li_2S$ to $Li_6PS_5Br$ and vice versa. Note that the star shape of the [7]Li resonance in $Li_6PS_5Br$ is the consequence of the 2D Fourier transform of its Lorentzian shape, a result of the high mobility of lithium ions within the $Li_6PS_5Br$ phase

Moreover, the similar interface and bulk resistances of mixtures I and III do not appear to be consistent with the much better capacities observed for mixture III compared to mixture I during galvanostatic charge/discharge cycling shown in Fig. 1b. Additionally, it appears unlikely that the bulk $Li_6PS_5Br$ conductivity is affected by cycling, and that annealing, going from mixture III to IV, increases both bulk and interfacial resistance. These inconsistent observations illustrate the difficulty in assessing the interface resistance in the complex bulk morphologies of these all-solid-state batteries[44].

Aiming at unambiguous quantification of the charge transfer kinetics over the $Li_6PS_5Br$–$Li_2S$ interface, and how this is affected by the battery preparation and cycling conditions, [7]Li-[7]Li 2D NMR exchange experiments are conducted for the mixtures I-IV, shown in Fig. 1a. 2D exchange NMR enables the measurement of spontaneous lithium-ion exchange between different lithium-ion environments[47, 49, 57], at present for the first time realized between a solid electrolyte and electrode material. These experiments provide selective and non-invasive quantification of the lithium-ion transport over the solid–solid

electrolyte–electrode interface in realistic solid-state cathode mixtures. [7]Li magic angle spinning (MAS) NMR spectra of the $Li_2S$–$Li_6PS_5Br$ mixtures I, II, and III are shown in Fig. 2a, e, i. MAS can average out anisotropic interactions that are described by rank-2 tensors, such as dipolar, first-order quadrupolar and chemical shift anisotropy, provided that the MAS frequency is larger than the interaction width. Compared to $Li_6PS_5Cl$, the [7]Li NMR resonance for $Li_6PS_5Br$ is shifted upfield caused by increased shielding of the lithium ions by the neighboring Br dopants. This results in a difference in chemical shift between Li in $Li_2S$ and in $Li_6PS_5Br$, which allows us to distinguish between lithium ions in $Li_6PS_5Br$ and $Li_2S$ phases, making it possible to conduct the 2D exchange NMR experiments. The 2D NMR spectra shown in Fig. 2 show that [7]Li in $Li_2S$ is represented by a broad homogeneous resonance, whereas [7]Li in $Li_6PS_5Br$ is represented by a star-shaped resonance. The latter is the consequence of the large lithium-ion mobility in the $Li_6PS_5Br$ solid electrolyte, which results in a Lorentzian line shape that upon 2D Fourier transformation results in the star-shaped NMR resonance observed.

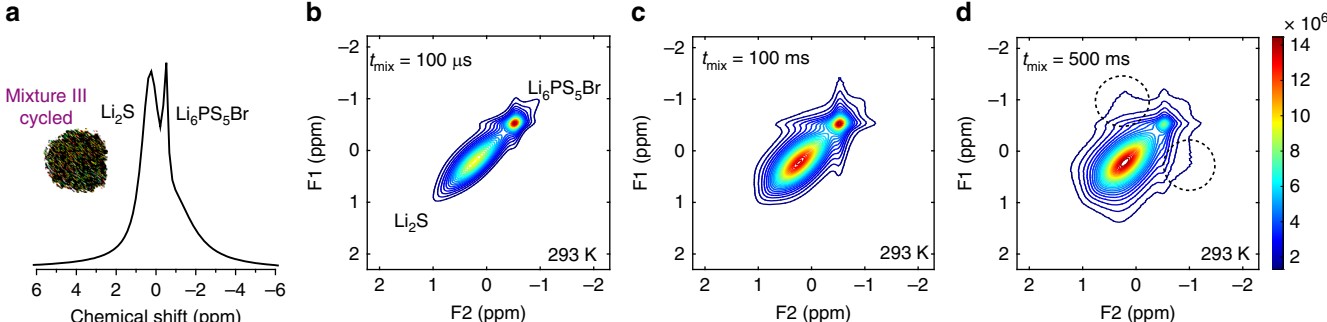

**Fig. 3** NMR measuring the spontaneous lithium-ion transport between the $Li_6PS_5Br$ solid electrolyte and the $Li_2S$ cathode after cycling. **a** One-dimensional (1D) $^7Li$ magic angle spinning (MAS) spectrum of the cycled $Li_6PS_5Br$–$Li_2S$ cathode mixture III. **b**-**d** Two-dimensional $^7Li$-$^7Li$ exchange spectra (2D-EXSY) measured under MAS at a resonance frequency of 330.39 MHz and spinning speed of 30 kHz at 298 K for the cycled mixture III at **b** $t_{mix} = 100$ μs, **c** $t_{mix} = 100$ ms, and **d** $t_{mix} = 500$ ms. Only after $t_{mix} = 500$ ms, off-diagonal exchange intensity is observed reflecting the lithium-ion exchange from $Li_2S$ to $Li_6PS_5Br$ and vice versa in the cycled cathode mixture III

2D exchange NMR effectively measures the spectrum of the $^7Li$ atoms at $t = 0$ s, then waits a "mixing time" $t_{mix}$, and subsequently measures the spectrum of the same $^7Li$ atoms again at $t = t_{mix}$. The results of such measurements are shown in Fig. 2. The signal occurring on the diagonal reflects the 1D NMR signal shown in Fig. 2a, e, i, which represents $^7Li$ atoms that have the same spectrum before and after $t_{mix}$. During $t_{mix}$, these lithium ions remained within the same material, either within $Li_6PS_5Br$ or within $Li_2S$. Off-diagonal intensity, clearly visible for mixture III at $t_{mix} = 10$ ms in Fig. 2k and strongly present at $t_{mix} = 100$ ms in Fig. 2l, represents lithium ions that at $t = 0$ were located in $Li_6PS_5Br$ and during $t = t_{mix}$ diffused to $Li_2S$ and vice versa. This off-diagonal intensity quantifies the amount of lithium ions that spontaneously moved between the electrode and the electrolyte during $t_{mix}$. Hence, by integrating the amount of off-diagonal intensity and dividing this by the intensity on the diagonal at $t_{mix} = 0$ and $t_{mix}$, we obtain the exchange current density, the amount of lithium ions that undergoes the charge transfer reaction between the $Li_6PS_5Br$ solid electrolyte and $Li_2S$ cathode. For the 2D spectrum in Fig. 2l, this results in that approximately 20% of the lithium ions moved from the $Li_6PS_5Br$ to the $Li_2S$ material in mixture III and vice versa within the mixing time $t_{mix} = 100$ ms. At $t_{mix} = 100$ μs, Fig. 2j, no off-diagonal signal is detected because this mixing time is too short for lithium ions to diffuse from $Li_6PS_5Br$ to $Li_2S$ material or vice versa. If the temperature is lowered to 248 K, the off-diagonal signal observed at large mixing times, shown in Supplementary Fig. 5, is much weaker compared to $t_{mix} = 100$ ms in Fig. 2l, because the lithium-ion motion between $Li_6PS_5Br$ and $Li_2S$ is frozen. In contrast at 348 K, the off-diagonal signal is stronger as shown in Supplementary Fig. 5 because of thermal activation of the lithium-ion diffusion between $Li_6PS_5Br$ and $Li_2S$. This indicates that the cross-peak intensities observed in Fig. 2 must be due to lithium-ion diffusion, and cannot arise from spin diffusion due to the presence of dipolar couplings (which are suppressed by MAS and also unlikely to cross the grain boundaries).

Comparing the 2D spectra at $t_{mix} = 100$ ms of mixtures I, II, and III in Fig. 2d, h, l, only mixture III displays evident off-diagonal intensity, which implies that only for mixture III there is significant lithium-ion transport between $Li_6PS_5Br$ and $Li_2S$ during 100 ms. From the exchanged amount of lithium ions, and taking into account the average crystallite sizes (23 nm for $Li_6PS_5Br$ and 38 nm for $Li_2S$ from XRD refinement and TEM, which appear to be close to the particle sizes from EDX, see Supplementary Figs. 1 and 2) the approximate exchange current density at room temperature can be calculated, amounting approximately to 1.0 mA cm$^{-2}$ for mixture III and less than

0.05 mA cm$^{-2}$ for mixtures I and II, which is small compared to the one in the liquid electrolyte lithium-ion batteries[49, 58]. The large increase in spontaneous lithium-ion transport between mixtures II and III indicates that reducing the $Li_2S$ particle size alone is not enough to provide significant lithium-ion transport over the $Li_6PS_5Br$–$Li_2S$ interface. Additionally, intimate mixing, here realized by high-speed ball milling as shown in Fig. 1a, appears essential, most likely because it creates more interfacial area. The improved lithium-ion transport over the interfaces going from mixture I and II toward III comes along with substantially better solid-state battery performance shown in Fig. 1b, indicating that facile lithium-ion transport is paramount for solid-state battery performance. Annealing mixture III at 150 °C, resulting in mixture IV, did not lead to a significantly different lithium-ion exchange compared to mixture III; hence, these mild annealing temperatures do not improve the interfaces with respect to the lithium-ion transport.

Figure 3 shows the impact of cycling on the spontaneous lithium-ion exchange between $Li_6PS_5Br$ and $Li_2S$ mixture III. At $t_{mix} = 100$ ms (Fig. 3c), no off-diagonal signal is observed, as opposed to the pristine uncycled mixture III that shows considerable lithium-ion exchange as shown in Fig. 2l. Only at $t_{mix} = 500$ ms in Fig. 3d, a weak signature of lithium-ion exchange between $Li_6PS_5Br$ and $Li_2S$ is observed. This proves that cycling changes the charge transfer over the interface considerably; leading to less facile lithium-ion transport over the solid–solid $Li_6PS_5Br$–$Li_2S$ interface.

In addition to the 2D exchange measurements, faster 1D $^7Li$–$^7Li$ exchange experiments under static conditions were performed to quantify the exchange as a function of $t_{mix}$ and temperature[24, 47, 49]. The much larger static spectral width of $^7Li$ in $Li_2S$ compared to that of $Li_6PS_5Br$, see Supplementary Fig. 6, is a consequence of the poor lithium-ion mobility in $Li_2S$ that is unable to average out the dipolar and first-order quadrupolar interactions. This makes it possible to selectively filter out the broad $Li_2S$ component using a $T_2$ filter. As a result, the repopulation of the $Li_2S$, through the transfer of magnetization carried by the $^7Li$ species diffusing from the $Li_6PS_5Br$ electrolyte back into $Li_2S$ is monitored as a function of the exchange time $t_{mix}$ at different temperatures, shown in Supplementary Fig. 7. When $t_{mix}$ exceeds 500 ms, the $T_1$ relaxation process dominates the decay of the total magnetization, which limits the evaluation of exchange to this timescale. In the present case, the $Li_2S$–$Li_6PS_5Br$ mixture, offers both a difference in chemical shift and a difference in line broadening, allowing quantification of lithium-ion exchange between these materials both by 2D exchange and by 1D exchange ($T_2$ filter) experiments, which

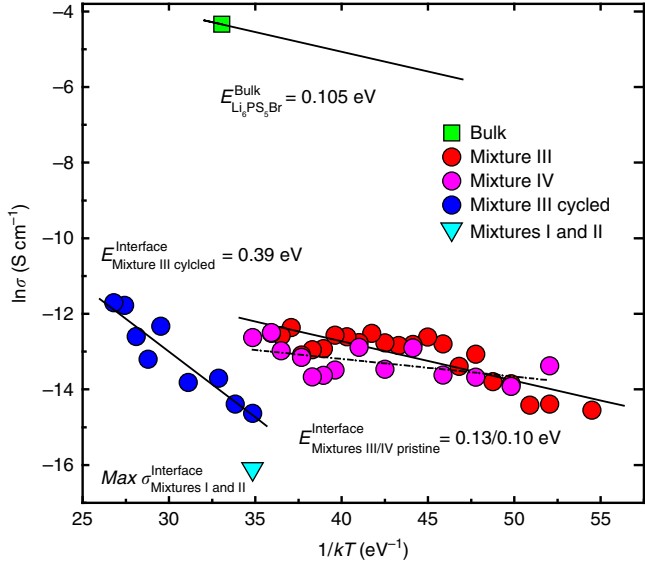

**Fig. 4** Comparison of the lithium-ion bulk and interface conductivities. Lithium-ion conductivity for bulk $Li_6PS_5Br$ determined by $^7Li$ solid-state NMR spin-lattice relaxation (SLR) experiments and the conductivity over the $Li_6PS_5Br$–$Li_2S$ interface from 2D/1D NMR exchange experiments for different cathode mixtures, I: micro $Li_2S$–$Li_6PS_5Br$, II: nano-$Li_2S$–$Li_6PS_5Br$, III: mixed nano-$Li_2S$–$Li_6PS_5Br$, IV: annealed mixed nano-$Li_2S$–$Li_6PS_5Br$ and V: mixture III cycled (see Fig. 1a)

are shown to be in excellent agreement in Supplementary Fig. 8. These two possibilities illustrate the versatility of NMR exchange for these systems. In general, the range of $^7Li$ chemical shifts in compounds is quite narrow, which in many cases may not provide enough contrast in chemical shift between the electrode and the solid electrolyte. However, typically solid electrolytes have larger conductivities compared to electrode materials, which in practice will provide for most combinations a difference in line broadening enabling the quantification of the exchange with 1D exchange ($T_2$ filter) experiments. Perhaps the largest practical restriction is the $T_1$ spin-lattice relaxation time of the materials, which determines the maximum exchange time that can be accessed. This is for instance a challenge for cathode materials having paramagnetic moments present. A potential strategy is to move toward $^6Li$, in general having larger values for $T_1$ because of the weaker spin-lattice coupling. However, the compromise is the lower abundance and sensitivity of $^6Li$, which increases measurement times. Therefore, we foresee that for most electrode–solid electrolyte combinations NMR exchange experiments will be able to quantify the lithium-ion transport over the interfaces, making this versatile, albeit, not straightforward approach to quantify the lithium-ion transport over the interfaces in all-solid-state batteries.

Quantification of exchange between the $Li_2S$ and $Li_6PS_5Br$ species was performed by fitting the growing $Li_2S$ signal (See in Supplementary Fig. 7) to a diffusion model, derived from Fick's law for diffusion as explained in detail in Supplementary Notes. From this, an average self-diffusion coefficient ($D$) as a function of temperature for the lithium-ion transfer over the electrolyte–electrode interface and the corresponding activation energy are determined shown in Supplementary Fig. 7. Here it is assumed that diffusion occurs from the center of a $Li_6PS_5Br$ solid electrolyte crystallite (with an average size of 23 nm) to the center of an $Li_2S$ particle (with an average size of 38 nm), where the average crystallite sizes are determined from XRD broadening. This assumes intimate mixing of $Li_6PS_5Br$ and $Li_2S$ on the length scale of these crystallite sizes, which is confirmed by the

distribution of Br and P determined by EDX mapping shown in Supplementary Fig. 2. This results in a self-diffusion coefficient for the $Li_6PS_5Br$–$Li_2S$ mixture III of $\sim 1 \times 10^{-11}$ cm$^2$ s$^{-1}$ at room temperature, both from the 2D and 1D exchange measurements. This is two orders of magnitude smaller than the self-diffusion coefficient for bulk diffusion, as determined from relaxation NMR ($10^{-9}$ cm$^2$ s$^{-1}$) both determined by Epp et al.[51] and at present (see Supplementary Fig. 9).

## Discussion

Using the Nernst–Einstein equation[59] and assuming no correlation effects[60], the conductivity can be calculated from both the bulk and interface self-diffusion coefficients. The resulting bulk conductivity, determined by the NMR relaxation experiments (see Supplementary Fig. 9), and the interface conductivities from the exchange experiments for mixtures I, II, III, and IV are shown as a function of temperature in Fig. 4. The activation energy of the bulk conductivity is obtained by fitting the high temperature slope of the $T_1$ relaxation, as shown in Supplementary Fig. 9. For mixtures I and II, no exchange was observed, but based on the maximum exchange time probed (250 ms), an upper limit of the conductivity is determined shown in Fig. 4. This illustrates that nanosizing $Li_2S$, and intimate mixing of nano-$Li_2S$ with $Li_6PS_5Br$ is essential to provide significant lithium-ion conductivity over the $Li_2S$–$Li_6PS_5Br$ interface. However, the conductivity over the interface appears several orders of magnitude smaller compared to the bulk $Li_6PS_5Br$ conductivity. The activation energy for lithium-ion transport over the $Li_2S$–$Li_6PS_5Br$ interface is slightly larger compared to that of the bulk $Li_6PS_5Br$ conductivity, indicating that in the pristine, uncycled mixture III the barrier for lithium-ion transport over the interface is small. However, the much smaller interface conductivity compared to the bulk conductivity suggests that there is poor wetting between $Li_2S$ and $Li_6PS_5Br$, hence there is little electrode–electrolyte contact area where the lithium-ion transport can take place.

Based on DFT calculations, it is expected that the argyrodite solid electrolytes are stable at the $Li_2S$ potential (~2.3 V)[40], and consequentially no redox instability should be expected in the pristine cathode mixture, which potentially could increase the interface impedance. This is consistent with the relatively small activation energy for lithium-ion transport over the interface for the uncycled mixture III, similar to that for the $Li_6PS_5Br$ bulk lithium-ion diffusion.

After two full charge–discharge cycles, between 0.62 and 4.12 V vs. Li$^+$/Li, the lithium-ion exchange is significantly lowered, as observed by comparing Figs. 2l and 3c. In Fig. 4, this results in a drop in conductivity of almost one order of magnitude (near room temperature). Additionally, the cycling raises the activation energy for lithium-ion transport over the $Li_2S$–$Li_6PS_5Br$ interface by a factor of three from 0.13 to 0.39 eV. This may be due to (1) large volumetric changes of $Li_2S$ upon charge and discharge causing contact loss between $Li_2S$ and $Li_6PS_5Br$ and (2) redox instabilities at the $Li_2S$–$Li_6PS_5Br$ interfaces leading to an interfacial layer that poses a higher barrier for lithium-ion transport. We anticipate that both play a role at the $Li_2S$–$Li_6PS_5Br$ interface, where (1) is responsible for the drop in interface conductivity because of loss of interfacial contact and (2) for the increase in activation energy due to an interfacial layer increasing the diffusion barrier. During cycling, the voltage was varied between 0.62 and 4.12 V vs. Li$^+$/Li, far outside the narrow electrochemical stability window predicted for these sulfide electrolytes[40, 41]. Recently, it was demonstrated that charging up to similar potentials results in oxidation of argyrodite $Li_6PS_5Cl$ toward elemental sulfur, lithium polysulfides, $P_2S_{x \leq 5}$ and LiCl[61]. Based on this, we suggest that the increase in activation energy is a result

**Table 1 Bulk and interface resistances of the mixtures derived from the conductivity in Fig. 4 determined from the NMR exchange experiments**

|  | NMR ($\Omega$)[a] |
|---|---|
| Mixture I |  |
| Bulk | 37 |
| Interface | >82 |
| Mixture II |  |
| Bulk | 37 |
| Interface | >82 |
| Mixture III |  |
| Bulk | 37 |
| Interface | 1.5 |
| Mixture IV |  |
| Bulk | – |
| Interface | 2.4 |
| Mixture III cycled |  |
| Bulk | 35 |
| Interface | 68 |

[a] Assuming for the bulk resistance an electrolyte thickness of 1600 μm (the thickness of the pellets tested by EIS) and a surface area of 0.78 cm$^2$ based on the diameter. And assuming an electrode–electrolyte interface area of 0.78 cm$^2$ (see text)

of the formation of an interfacial layer of similar oxidation products from the $Li_6PS_5Br$ solid electrolyte that develop during the first cycles. Additionally, sulfide electrolytes have been reported to act as active materials during charging[62]. Oxidation of $Li_6PS_5Br$ may be responsible for the change in the $T_1$ in the cycled mixture III compared to the pristine material, both shown in Supplementary Fig. 9. However, it appeared difficult to obtain accurate fits for the $T_1$ of $Li_6PS_5Br$ due to the presence of $Li_2S$ in the cycled mixtures. To investigate the influence of oxidation of $Li_6PS_5Br$ on the $T_1$ further in the absence of $Li_2S$, $Li_6PS_5Br$ was charged by using it as cathode (by mixing with carbon). The result shown in Supplementary Fig. 9 shows that $T_1$ is hardly affected by charging, indicating that the bulk of the $Li_6PS_5Br$ material remains intact not excluding that the surface may be oxidized.

To assess the role of the conductivity over the $Li_2S$–$Li_6PS_5Br$ interface in the all-solid-state batteries, the resistance due to both the $Li_6PS_5Br$ solid electrolyte and the $Li_2S$–$Li_6PS_5Br$ interface is approximated from the NMR results. The $Li_6PS_5Br$ bulk conductivity is determined from the $T_1$ relaxation experiments (Supplementary Fig. 9). In combination with the solid electrolyte pellet thickness, $l$, and surface area, $A$, as used for the electrochemical impedance spectroscopy (see Supplementary Figs. 4 and 9) and the cycling performance shown in Fig. 1b, the bulk solid electrolyte resistance $R$ is calculated with $R = l/(A\sigma)$ and reported in Table 1. To calculate the $Li_2S$–$Li_6PS_5Br$ interface resistance, it is assumed that interface thickness, $l$, equals the average distance between the solid electrolyte and the electrode, which was also used to calculate the diffusion coefficient from the NMR exchange experiments. The interface area, $A$, between $Li_2S$ and $Li_6PS_5Br$ in the cathode mixtures is difficult to estimate, and therefore for simplicity we assume the interface area to be equal to the area of the solid electrolyte pellet, to make at least a qualitative comparison possible, which results in the interface resistance determined from the NMR exchange experiments in Table 1.

The resistances in Table 1, estimated from the exchange NMR, provide a consistent picture of the role of the electrode–electrolyte interface on the solid-state battery performance of the different mixtures shown in Fig. 1b. The results for the bulk and interface resistance obtained by electrochemical impedance spectroscopy (EIS) reported in Supplementary Table 1 suggest that the battery performance of mixtures I, II, and III should be similar. In contrast, the NMR exchange results in Table 1 indicate that the

high interfacial resistances for mixtures I and II are responsible for the poor capacities during galvanostatic cycling observed in Fig. 1b. This can be explained by a combination of both $Li_2S$ nanosizing and mixing with the $Li_6PS_5Br$, resulting in much more interfacial area between the $Li_2S$ electrode and the $Li_6PS_5Br$ solid electrolyte. Table 1 indicates that in the pristine cathode mixture III, the interfacial resistance is relatively small compared to the bulk resistance of a 1600 μm thick electrolyte. In practice, a solid electrolyte thickness of 100 μm is more realistic to achieve high-energy density and power density, which would result in a bulk resistance of just 1.9 Ω for the pristine cathode mixture III, comparable to the interface resistance of 1.5 Ω. Just two galvanostatic charge/discharge cycles increase the interface resistance to 68 Ω, illustrating that the overpotential during galvanostatic cycling will be dominated by the lithium-ion transport over interface between the $Li_2S$ cathode and the $Li_6PS_5Br$ solid electrolyte. As discussed above, this is most likely a consequence of both contact loss due to the large volumetric changes of the cathode and an interfacial barrier arising from the solid electrolyte oxidation and/or reduction.

Using the $Li_2S$–$Li_6PS_5Br$ solid-state battery as an example, the present experimental results demonstrate that lithium-ion interfacial transport over the electrode–electrolyte interfaces is the major bottleneck to lithium-ion transport through all-solid-state batteries. Both the preparation conditions and battery cycling affect interfacial transport considerably. Therefore, realizing high-energy density all-solid-state batteries will require interface design to prevent the large increase in impedance during cycling, where in particular volumetric changes and redox instabilities appear responsible. This work demonstrates the ability of exchange NMR between distinguishable lithium-ion sites in the electrode and the solid electrolyte to quantify unambiguously the amount and timescale of lithium-ion transport over the solid electrolyte–electrode interface in bulk solid-state batteries. Thereby this approach may be a valuable support to the development of interface design strategies necessary for future high-performance all-solid-state batteries.

## Methods

**Solid electrolyte and cathode mixture preparation**. Reagent-grade $Li_2S$ (99.98%, Sigma-Aldrich), $P_2S_5$ (99%, Sigma-Aldrich), and LiBr (99.0%, Sigma-Aldrich) crystalline powders were used as raw materials. The required amount of starting materials according to the molar ratios were ball milled in a WC-coated (inner) stainless steel jar with 10 WC balls (8 g per ball) filled in an argon-filled glove box ($H_2O$, $O_2$ < 0.3 ppm) because of the reactivity with oxygen and moisture. The total weight of the mixture was almost 2.0 g in the jar and the ball milling rotation speed was fixed at 600 rpm for 15 h. After the ball milling process, the mixture was sealed in a quartz tube and annealed at 300 °C for 5 h to obtain the final $Li_6PS_5Br$ powder. The pristine $Li_2S$–$Li_6PS_5Br$ mixtures I–IV used in this work were prepared as follows; for mixture I, commercial $Li_2S$ (99.98%, Sigma-Aldrich) and ball-milled $Li_6PS_5Br$ (450 rpm for 4 h) were mixed and milled with a rotation speed of 110 rpm for 1 h. For mixture II, nano-$Li_2S$ (obtained by milling the commercial $Li_2S$ with rotation speed of 500 rpm for 4 h) was mixed with the same $Li_6PS_5Br$ using a speed of 110 rpm for 1 h. For mixture III, the above nano-$Li_2S$ was milled with $Li_6PS_5Br$ with a rotation speed of 500 rpm for 1 h. All of those mixtures were pressed into pellets with a diameter of 10 mm and then crushed into small pieces for the final ion exchange NMR experiments. Mixture IV was prepared by annealing a pellet pressed from mixture III at 150 °C for 3 h. The weight ratio of $Li_2S$ and $Li_6PS_5Br$ in all four mixtures was fixed to 1:1.

**Impedance spectroscopy, XRD, TEM, and EDX material characterization**. Ionic conductivities of the annealed $Li_6PS_5Br$ solid electrolyte were measured by pelletizing the powder to a 10 mm diameter. Stainless steel disks were attached to both faces of the pellet. AC impedance measurements were performed for the cell by an Autolab (PGSTAT302N) in the frequency range of 0.1 Hz–1 MHz with an applied voltage of 0.05 V.

Powder X-ray diffraction (XRD) patterns were collected over a two-theta range of 10–80° to identify the crystalline phases of the prepared materials using $Cu_{K\alpha}$ X-rays (1.5406 Å at 45 kV and 40 mA) on an X'Pert Pro X-ray diffractometer (PANalytical). To prevent reaction with moisture and oxygen, the powder

materials were sealed in an airtight XRD sample holder in an argon-filled glove box.

For the TEM and energy dispersive X-ray (STEM-EDX) investigations, a suspension in dry ethanol was prepared, which was drop casted onto a standard gold grid with a holy carbon film, inside an argon-filled glove box. To prevent any contact with air TEM grids with the sample were loaded into a custom-made vacuum transfer TEM holder. TEM measurements were carried out in a FEI-Tecnai operating at 200 kV.

**Solid-state lithium battery preparation and electrochemical impedance spectroscopy**. Laboratory scale solid-state Li–S batteries were fabricated in the following manner. Each of the pristine mixtures (I–IV) was milled with super P with a weight ratio of 4:1 using a rotation speed of 110 rpm for 1 h to obtain the final cathode mixture. Then, a two-layer pellet ($d = 10$ mm), consisting of 12 mg the described cathode mixture and 88 mg of the $Li_6PS_5Br$ electrolyte, was prepared by pressing them together under 6 tons per $cm^2$. After that, a piece of In foil was attached to the other side. Then, the whole triple-pellet was pressed under 2 tons per $cm^2$ of pressure for 30 s. The assembled cells were charged and discharged under a current density of 0.064 mA $cm^{-2}$ between 0 and 3.5 V vs. In to evaluate their electrochemical performances. In addition, the cycled mixture III was obtained by collecting the cathode mixture III after two full charge–discharge cycles in the solid-state cell. The capacities obtained were normalized by the weight of $Li_2S$ in the cathode mixture. Electrochemical impedance spectrometry (EIS) measurements were performed with an Autolab PGSTAT302N before and after several charge–discharge cycles in the frequency range of 0.1 Hz–1 MHz with an applied voltage of 0.05 V.

**Solid-state [7]Li NMR measurements**. [7]Li static and MAS solid-state NMR measurements were performed on a Chemagnetics 400 Infinity spectrometer ($B_0 = 9.4$ T, 155.506 MHz for [7]Li). The $\pi/2$ pulse length was determined to be 3.2 μs with an RF field strength of 84 kHz for the static and 2.3 μs with an RF field strength of 120 kHz for the MAS measurements. Chemical shifts were referenced with respect to a 0.1 M LiCl solution. For the static NMR measurements, the air sensitive $Li_6PS_5Br$ solid electrolyte sample and the $Li_2S$ electrode–$Li_6PS_5Br$ solid electrolyte mixtures were sealed in custom-made Teflon tubes in an argon-filled glove box ($H_2O$, $O_2 < 0.3$ ppm). Variable temperature one-dimensional (1D) exchange measurements were performed using a 5 mm static goniometer probe from 213 to 433 K. $T_1$ relaxation times were additionally determined at various temperatures using a saturation recovery experiment. The pulse sequence used has been described in detail elsewhere with the appropriate phase cycle for cancellation of direct magnetization that may occur after $T_1$ relaxation[47, 49]. Briefly, the sequence consists of $\pi/2$, $\tau$, $\pi$, $\tau$, $-\pi/2$, $t_{mix}$, $+\pi/2$, acquisition. An echo time $\tau$ ranging from 200 to 800 μs was utilized to preserve the intensity of the narrow $Li_6PS_5Br$ resonance and filter out the broad $Li_2S$ resonance, effectively functioning as a $T_2$ filter. These 1D exchange experiments were performed for a range of mixing times, $t_{mix}$, to follow the spontaneous equilibrium exchange of Li between the $Li_6PS_5Br$ and $Li_2S$ phases. Lithium-ion exchange between the $Li_6PS_5Br$ and $Li_2S$ phases for mixtures I–IV was also measured under MAS with a 3.2 mm T3 MAS probe at a spinning speed of 20 kHz with two-dimensional rotor synchronized exchange spectroscopy (2D-EXSY) experiments performed at 348, 298, and 248 K at various mixing times[63, 64]. 2D exchange experiments for cycled mixture III were performed on a Varian VNMRS 850 MHz spectrometer ($B_0 = 20$ T, 330.2 MHz for [7]Li) using a triple resonance 1.6 mm Varian T3MAS probe at 30 kHz MAS at 298 K. The $\pi/2$ pulse length was determined to be 2.2 μs with an RF field strength of 130 kHz. All 2D spectra consist of 16 scans for each of the 200 transients, each transient incremented by 200 μs with a recycle delay of up to 5 s.

**Data availability**. The data supporting the findings of this study are available from the authors on reasonable request.

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

## Acknowledgements

The research leading to these results has received funding from the European Research Council under the European Union's Seventh Framework Programme (FP/2007-2013)/ ERC Grant agreement no. 307161 to M.W. Technical support from Frans G.B. Ooms, Michel Steenvoorden, Jouke Heringa and Bert Zwart is greatly acknowledged. Support from the Dutch organization of scientific research (NWO) for the solid-state NMR facility for advanced materials science in Nijmegen is gratefully acknowledged. The technical assistance of Hans Janssen, Gerrit Janssen, and Jan Schoonbrood is gratefully acknowledged.

## Author contributions

C.Y. made the samples. S.G., E.R.H.V.E., and C.Y. performed the NMR experiments, S.G. analyzed the experiments. M.W., S.G., and C.Y. designed the experiments. M.W. wrote the manuscript. C.Y. and H.W. performed and analyzed the impedance spectroscopy. S.B. and Z.L. performed the TEM/EDX/SEM experiments.

## Additional information

**Competing interests:** The authors declare no competing financial interests.

