## [Peer Review File · Nature Communications]

Reviewers' comments:

Reviewer #1 (Remarks to the Author):

Opinion from the reviewer:

Publish elsewhere

Comments:

The manuscript mainly reports the exchange NMR for distinguishing Li-ion sites in electrodes and solid-state electrolytes. Using the Li₂S-Li₆PS₅Br solid-state battery as an example, the authors claim that this method can quantify the Li-ion transportation over the electrolyte interface. I should say the 2D Li exchange NMR is a good supplement to existing characterization tools to study Li-ion transportation in electrodes and electrolytes. However, it is found that the role of the approach in this study is overstated. Additionally, the study on an approach for interface resistance in batteries is too specialized to meet broad interest for Nat. Commun. I would suggest the authors submit this work to a more specialized journal after making the following revisions.

1. The authors concluded that the exchange NMR can “quantify unambiguously the transport over the solid electrolyte–electrode interface, which is impossible by using any other technique up to date”. In fact, the interface resistance (in Table 2) was estimated roughly based on many assumptions, including the conductivity, particle size, interface area, etc. Therefore, such a statement is to some extent a hype. The authors should make an appropriate assessment of this method.
2. The discussion of EIS in Figure S13 and Table 1 is questionable. The authors used a doubtful equivalent circuit R(RQ)Q, which cannot actually fit the EIS in Figure S13. The authors should adopt a better equivalent circuit, for example, R(QR)(Q(RW)), to correctly fit the bulk and interface resistances. Because the EIS fitting in this paper is questionable, the conclusion on the ability of EIS in assessing the interfacial resistance is not very fair. The authors should better fit the EIS data, draw the Nyquist plot and fitted curve clearly, and assess the EIS method based on correct fitting results.
3. The authors are suggested to include some discussion on the versatility of this method. Although the Li sites in Li₂S and Li₆PS₅Br can be distinguished by NMR, is it easy to tell Li sites in other electrode and electrolyte materials, such as NMC and LISICON?
4. The crystalline size calculated via Rietveld refinement is not equal to particle size. The actual particle size should be much larger than the crystalline grains. The particle size should be measured by SEM or laser diffraction.
5. In the 2D-EXSY diagrams, intensity bars are suggested to show the contrast from blue to red lines.

Reviewer #2 (Remarks to the Author):

This study reports using the diffusion NMR approach as a non-invasive tool to evaluate the interfacial contact between Li₂S electrode and Li₆PS₅Br electrolyte in all-solid-state batteries. It showed that by nanosizing Li₂S, the electrode-electrolyte interfacial contact was improved, interfacial conductivity was reduced, and battery cycling performance was made better. This study addresses an important issue in the field of solid-state electrolytes for the next generation rechargeable Li-ion batteries, i.e., how to reduce interfacial impedance. Overall, the quality is suitable for this prestigious journal. Improvement is possible if the authors could address/clarify the following questions.

- A mixed-up term. The diffusivity DEXSY measured from EXSY NMR is for nuclear polarization transfer rate (Li ions do not have to physically move, as a matter of fact, the movement of Li ions will reduce D), although it is very relevant to evaluate the interface contact, it is very different from D' which characterizes how fast for Li ions physically diffuse from Li₂S to Li₆PS₅Br.
- The paper has a few hypothesis regarding the increase of interfacial impedance after

electrochemical cycling. These hypotheses can be tested with TEM (volume expansion hypothesis) and XPS (SEI) formation. In addition, from the VT T1 relaxation measurements, it seems the bulk Li₆PS₅Br also degraded as a result of the cycling.

– Is it possible to determine the activation energy of the bulk Li₆PS₅Br with more than one data point in Fig. 4?

Reviewer #3 (Remarks to the Author):

Reviewer's comment for NCOMMS-17-05273

This manuscript reports two-dimensional Li-ion exchange-NMR method for investigating Li-ion interfacial transport in all-solid-state batteries. The NMR technique successfully distinguishes interface Li ions from bulk ones (in Li₆PS₅Br solid electrolyte and Li₂S active material). Activation energy for interfacial ion-transport between Li₆PS₅Br and Li₂S was determined by the NMR technique, and it increased after charge-discharge cycling. This technique sounds useful for investigating Li⁺ ion transport ability at the electrode-electrolyte interface. However, several points have not been understood yet and some revisions based on the following comments are needed.

The comments are as follows:

1. Activation energy for ion transport at the electrode-electrolyte interface as shown in Fig. 4 and Fig. S18 can be determined by not only 2D-NMR but also 1D-NMR. 1D-NMR technique is simpler and used widely, and thus the merit of the use of 2D-NMR in addition to 1D-NMR should be emphasized in the revised manuscript.

2. A higher charge plateau at 2.8 V vs. Li appears in all-solid-state cell with the mixture III as shown in Fig. S12. This plateau would be attributable to Li⁺ de-intercalation from solid electrolytes. Sulfide electrolytes such as Li₃PS₄ are reported to act as an active material by high-energy ball-milling with carbon additives [A]. The mixture III has wide contact area between electrode and electrolyte, and thus the Li₆PS₅Br electrolyte is partially used as an active material. How about the effect of de-intercalated Li₆PS₅Br on the NMR spectra?

[A] T. Hakari et al., J. Power Sources, 293 (2015) 721.

3. Charge-discharge profile and cycle performance of mixture IV should be added to Fig 1b and Fig. S12.

4. Cycle performance of all-solid-state batteries using the mixture III is shown in Fig. 1. Capacity fading is observed and this charge-discharge performance should be compared with those reported in solid-state batteries using Li₂S-electrolyte positive electrodes.

5. As shown in Table 1, a bulk resistance of the mixture III is increased by charge-discharge cycles. Why was the resistance of an electrolyte separator layer increased? A detailed explanation is needed. The bulk and interface resistances in Table 2 are different from those in Table 1. The difference is discussed in the revised manuscript.

6. The authors mentioned that about 20% Li-ions was moved from the Li₆PS₅Br to the Li₂S in mixture

III in Fig. 2I. Quantitative determination of Li-ion amount is important, but calculation process has not been described. Detailed explanation should be added in the revised manuscript.

REVIEWERS' COMMENTS:

Reviewer #1 (Remarks to the Author):

Publish as is. No further revision needed.

Reviewer #2 (Remarks to the Author):

The authors have adequately addressed the reviewer's questions/comments based on further experiments and previous reports. The reviewer considers the revised manuscript suitable for publication in Nature Communications.

Reviewer #3 (Remarks to the Author):

The authors have replied to all the reviewer's comments and the revisions sound reasonable.

Reviewer #1 (Remarks to the Author):

Opinion from the reviewer:

Publish elsewhere

Comments:

The manuscript mainly reports the exchange NMR for distinguishing Li-ion sites in electrodes and solid-state electrolytes. Using the Li₂S-Li₆PS₅Br solid-state battery as an example, the authors claim that this method can quantify the Li-ion transportation over the electrolyte interface. I should say the 2D Li exchange NMR is a good supplement to existing characterization tools to study Li-ion transportation in electrodes and electrolytes. However, it is found that the role of the approach in this study is overstated. Additionally, the study on an approach for interface resistance in batteries is too specialized to meet broad interest for Nat. Commun. I would suggest the authors submit this work to a more specialized journal after making the following revisions.

1. The authors concluded that the exchange NMR can “quantify unambiguously the transport over the solid electrolyte–electrode interface, which is impossible by using any other technique up to date”. In fact, the interface resistance (in Table 2) was estimated roughly based on many assumptions, including the conductivity, particle size, interface area, etc. Therefore, such a statement is to some extent a hype. The authors should make an appropriate assessment of this method.

We fully agree with the reviewer that we have to be careful with the statement “quantify unambiguously the transport over the solid electrolyte-electrode interface, which is impossible by using any other technique up to date”. In this sentence we were referring to the absolute amount of exchange and the time scale. This is unambiguously quantified by the exchange NMR, as it provides the amount of Li-exchanged and the timescale, see also comment 6 of Reviewer #3. We agree that the conductivity and the resistance required a number of assumptions which have been stated in the manuscript. To be precise, obtaining the diffusion coefficient requires the average particle size (as stated in the manuscript), and calculating the conductivity from that using the Nernst-Einstein equation assumes no correlation effects (as mentioned in the manuscript). As advised by the reviewer in comment 4, in addition to diffraction, we have provided TEM pictures, as well as EDX maps to confirm the crystallite particle size and the intimate mixing of both Li₆PS₅Br and Li₂S. Thereby, we believe it is valid to state that the conductivity is obtained from the NMR without critical assumptions. However, it is true that for calculation of the resistance we need to make an assumption on the interface area (as discussed in the manuscript).

Revisions made:

We have revised the line as follows: “quantify unambiguously the amount and timescale of Li-ion transport over the solid electrolyte-electrode interface”.

In addition, we have added in the manuscript how this is unambiguously quantified, thereby also answering comment 6 of Reviewer #3.

Both TEM pictures and EDX maps, demonstrating the crystallite particle size, and intimate mixing of both Li₆PS₅Br and Li₂S are added to the supplementary information (Figure SI2).

Finally, we added an assessment of the exchange NMR method to the manuscript, see also comment 3 of Reviewer #1, restating the assumptions to make this very transparent to the reader.

2. The discussion of EIS in Figure SI3 and Table 1 is questionable. The authors used a doubtful equivalent circuit $R(RQ)Q$, which cannot actually fit the EIS in Figure SI3. The authors should adopt a better equivalent circuit, for example, $R(QR)(Q(RW))$, to correctly fit the bulk and interface resistances. Because the EIS fitting in this paper is questionable, the conclusion on the ability of EIS in assessing the interfacial resistance is not very fair. The authors should better fit the EIS data, draw the Nyquist plot and fitted curve clearly, and assess the EIS method based on correct fitting results.

We thank the Reviewer for this important comment and useful suggestion. Our starting point was the work of Huang, B. et al. J. Power Sources 284, 206-211 (2015) as this equivalent circuit, $R_1(R_2Q_2)Q_3$ that suggested for these type of all-solid-state batteries. Following the suggestion of the reviewer we have fitted the EIS data with the $R_1(Q_2R_2)(Q_3(R_3W))$ equivalent circuit proposed by the reviewer. Additionally we have fitted the EIS data with a third equivalent circuit: $R_1(R_2Q_2)(R_3Q_3)Q_4$. However, all three equivalent circuits, do not result in very good fit quality, in particular at high frequencies as shown in the new Figure SI4 Part I showing the fits for the three equivalent circuits for the four pristine and cycled mixtures. A number of the fitted resistances show a standard deviation exceeding 100%, and as a consequence this does not allow reasonable interpretation.

Therefore we conclude at this stage, also after discussing with several impedance experts in the field, that there appears to be no rational simple equivalent circuit available for solid-state batteries that are able to provide high quality fits. Because in this case we are mainly interested in the values of R_1 and R_2 (representing the solid electrolyte bulk resistance and the interface resistance) for comparison with the exchange NMR results, we followed the following approach: The uncycled mixtures were best fit by the most simple $R_1(R_2Q_2)Q_3$ equivalent circuit, and because of the appearance of a third semi-circle, the cycled mixtures were best fitted by the $R_1(R_2Q_2)(R_3Q_3)Q_4$ equivalent circuit. A possible explanation for the third semi-circle is the formation of an interface at the In anode. To gain the best fits for R_1 and R_2 only the high frequency part of the spectrum was used for the fits, shown in Figure SI4 Part II. The resulting fit values are marked green in Table SI1. These values do not display a systematic variation for the different mixtures that can be related with the battery cycling performance, which supports our conclusion that EIS in this case appears unable to capture the large increase in interfacial resistance, which is observed with the exchange NMR.

Revisions made:

The EIS data was fitted with the $R(QR)(Q(RW))$ circuit proposed by the Reviewer and an additional $R_1(R_2Q_2)(R_3Q_3)Q_4$ circuit. The fits of these three equivalent models, and the resulting resistances are added to the supplementary information, Figure SI4 Part I and Part II and Table SI1. In the manuscript we have mentioned the difficulty of all three models to accurately fit the EIS data, motivating the use of the most simple model to determine the solid electrolyte resistance and the interface resistance. We have moved Table 1 to the supporting information (Table SI1). In the captions of Figure SI4 and Table SI1, the EIS fitting strategy is motivated. As requested we also plotted the Nyquist plot including the fits in Figure SI4, making it easier to evaluate the EIS data and fits.

3. The authors are suggested to include some discussion on the versatility of this method. Although the Li sites in Li_2S and Li_6PS_5Br can be distinguished by NMR, is it easy to tell Li sites in other electrode and electrolyte materials, such as NMC and LISICON?

We thank the reviewer for this very useful suggestion. We are currently working on exchange experiments between argyrodites (and other sulfide electrolytes) and $LiCoO_2$ and NCM, as well as between LISICON and several cathodes. For all these combinations exchange experiments are able to

quantify the exchange, however, each combination requires a slightly different NMR approach. Below we have described the possibilities, illustrating the versatility of this method.

Revisions made:

The following discussion/assessment on the versatility of this method is added to the manuscript:

In the present case, $\text{Li}_2\text{S-Li}_6\text{PS}_5\text{Br}$, offers both a difference in chemical shift and a difference in line broadening allowing to quantify exchange both by 2D exchange and by the 1D exchange (T_2 filter) experiments, which are shown to be in good agreement (see Figure SI7 in the supporting information). These two possibilities illustrate the versatility of the NMR exchange method. In general the range of ^7Li chemical shifts for various compounds is quite narrow, which in many cases may not provide enough difference in chemical shift between the electrode and the solid electrolyte. However, typically solid electrolytes have larger conductivities compared to electrode materials, which in practice will provide for most combinations a difference in line broadening enabling the quantification of the exchange by 1D exchange (T_2 filter) experiments. Perhaps the largest practical restriction is the T_1 spin-lattice relaxation time of the materials, which determines the maximum exchange time that can be accessed. This is for instance a challenge for cathode materials having paramagnetic moments present. A potential strategy is to move towards ^6Li , in general having larger values for T_1 because of the weaker spin-lattice coupling. However, the compromise is the lower abundance and sensitivity of ^6Li , which increases measurement times. Therefore, we foresee that for most electrode - solid electrolyte combinations NMR exchange experiments will be able to quantify the Li-ion transport over the interfaces, making this versatile, albeit, not straightforward approach to quantify the Li-ion transport over the interfaces in all-solid-state batteries.

4. The crystalline size calculated via Rietveld refinement is not equal to particle size. The actual particle size should be much larger than the crystalline grains. The particle size should be measured by SEM or laser diffraction.

We thank the reviewer for this important comment, which deserves more attention. We agree that the crystallite size from diffraction is not necessarily equal to the particle size as it may be agglomerates, and therefore we have investigated the mixtures in more detail. SEM was not able to determine the particle size and length scale of mixing and laser diffraction appeared impossible for these mixtures. Finally, we have performed TEM and EDX, results of which are added to the supplementary information as Figure SI2. From the TEM crystalline regions are observed that have a size in the order of that resulting from Rietveld refinement. However, the disorder in the mixtures makes it very hard to estimate particle sizes. As the Reviewer indicates, the relevant size is the particle size, or actually the length scale of the mixing of Li_2S and $\text{Li}_6\text{PS}_5\text{Br}$, as this length scale is used to determine the diffusion coefficient from the exchange NMR experiments (see our answer to comment 1). Because this appeared impossible to determine from TEM, we measured EDX maps of the mixtures, also added to the supplementary information in Figure SI2. Although difficult to quantify accurately, the EDX maps show mixing of Li_2S and $\text{Li}_6\text{PS}_5\text{Br}$ on the length scale consistent with the crystallite sizes as determined from XRD. Therefore we conclude that the crystallite size from Rietveld refinement is the best quantitative length scale to be used to determination of the diffusion coefficients (and conductivities) from the NMR exchange data. We do realize this may introduce a systematic error, but the relative changes in the diffusion coefficients can be trusted.

Revisions made:

Both TEM pictures and EDX maps, demonstrating the crystallite particle size, and intimate mixing of both $\text{Li}_6\text{PS}_5\text{Br}$ and Li_2S are added to the supplementary information Figure S12.

5. In the 2D-EXSY diagrams, intensity bars are suggested to show the contrast from blue to red lines.

We agree, we have revised Figure 2 and 3 by adding an intensity bar that quantifies the color scheme.

Reviewer #2 (Remarks to the Author):

This study reports using the diffusion NMR approach as a non-invasive tool to evaluate the interfacial contact between Li_2S electrode and $\text{Li}_6\text{PS}_5\text{Br}$ electrolyte in all-solid-state batteries. It showed that by nanosizing Li_2S , the electrode-electrolyte interfacial contact was improved, interfacial conductivity was reduced, and battery cycling performance was made better. This study addresses an important issue in the field of solid-state electrolytes for the next generation rechargeable Li-ion batteries, i.e., how to reduce interfacial impedance. Overall, the quality is suitable for this prestigious journal. Improvement is possible if the authors could address/clarify the following questions.

1. A mixed-up term. The diffusivity DEXSY measured from EXSY NMR is for nuclear polarization transfer rate (Li ions do not have to physically move, as a matter of fact, the movement of Li ions will reduce D), although it is very relevant to evaluate the interface contact, it is very different from D' which characterizes how fast for Li ions physically diffuse from Li_2S to $\text{Li}_6\text{PS}_5\text{Br}$.

We thank the reviewer for this comment. As far as we can find back in literature and text books EXSY NMR in general is used to indicate exchange NMR spectroscopy which refers to diffusion of polarization, either by spin diffusion or by physical diffusion of for instance Li-ions. The Reviewer correctly indicates that this may also be used to quantify spin waves, which does not require physical motion of the Li-ions. In the present case we prove there is no transport of polarization through spin waves, because the measured exchange can be frozen at temperatures far above 10 Kelvin. Moreover, spin diffusion is unlikely to occur over grain boundaries.

Revision made:

To avoid confusion we have avoided the term EXSY and replaced it by 2D-exchange NMR.

2. The paper has a few hypothesis regarding the increase of interfacial impedance after electrochemical cycling. These hypotheses can be tested with TEM (volume expansion hypothesis) and XPS (SEI) formation. In addition, from the VT T_1 relaxation measurements, it seems the bulk $\text{Li}_6\text{PS}_5\text{Br}$ also degraded as a result of the cycling.

We thank the reviewer for this valuable suggestion. A very recent publication appeared using TEM and XPS reports results in line with the hypothesized side reactions that provide a rationale for the increased impedance [Auvergniot et al. Chemistry of Materials 2017 29 (9), 3883-3890], which is now added to the discussion in the manuscript.

We agree with the Reviewer that the VT T_1 experiments may indicate that the cycled $\text{Li}_6\text{PS}_5\text{Br}$ has degraded. However, the T_1 of the $\text{Li}_6\text{PS}_5\text{Br}$ is difficult to determine from the cycled $\text{Li}_6\text{PS}_5\text{Br}$ - Li_2S mixture due to the T_1 of the cycled Li_2S . For this reason we performed additional experiments to investigate the influence of cycling on the $\text{Li}_6\text{PS}_5\text{Br}$, without the presence of Li_2S , by charging $\text{Li}_6\text{PS}_5\text{Br}$ as cathode (by mixing it with carbon black). This more accurate data is added to Figure S19, showing that charging hardly affects the T_1 , most likely because only the surface of the $\text{Li}_6\text{PS}_5\text{Br}$ material may

be oxidized, also consistent with TEM and XPS [Auvergniot et al. *Chemistry of Materials* **2017** 29 (9), 3883-3890]. We have also performed additional TEM experiments on cycled materials, see new Figure SI2, however the large disorder did not result in conclusive evidence the impact of the volumetric changes. Therefore, the increased impedance based on volumetric changes remains a hypothesis, although loss of contact associated with the volumetric changes is consistent with the observed lowering of the conductivity upon cycling.

Revisions made:

The reference regarding TEM and XPS on argyrodites is added and discussed, indicating that the oxidation of argyrodites to elemental sulfur, lithium polysulfides, P_2S_5 , phosphates, and LiCl is responsible for the increase in the impedance [Auvergniot et al. *Chemistry of Materials* **2017** 29 (9), 3883-3890].

We have added VT T_1 NMR results of charged (oxidized) Li_6PS_5Br to the supplementary information Figure SI9, and added a short discussion on this in the manuscript, in relation the above reference, giving a rational for the observed increase in interface impedance.

TEM (and EDX) images have been added to Figure SI2.

3. Is it possible to determine the activation energy of the bulk Li_6PS_5Br with more than one data point in Fig. 4?

We thank the reviewer for this comment. Actually the activation energy is from the VT T_1 experiment shown in Supplementary Figure SI8. The high temperature slope quantifies the activation energy for Li-ion diffusion, hence the activation is the result of approximately 15 points where T_1 was measured at 15 different temperatures.

Revisions made:

To avoid confusion, we have added the fit in Figure SI8, and added a sentence in the manuscript stating how the activation energy was obtained.

Reviewer #3 (Remarks to the Author):

Reviewer's comment for NCOMMS-17-05273

This manuscript reports two-dimensional Li-ion exchange-NMR method for investigating Li-ion interfacial transport in all-solid-state batteries. The NMR technique successfully distinguishes interface Li ions from bulk ones (in Li_6PS_5Br solid electrolyte and Li_2S active material). Activation energy for interfacial ion-transport between Li_6PS_5Br and Li_2S was determined by the NMR technique, and it increased after charge-discharge cycling. This technique sounds useful for investigating Li^+ ion transport ability at the electrode-electrolyte interface. However, several points have not been understood yet and some revisions based on the following comments are needed.

The comments are as follows:

1. Activation energy for ion transport at the electrode-electrolyte interface as shown in Fig. 4 and Fig. SI8 can be determined by not only 2D-NMR but also 1D-NMR. 1D-NMR technique is simpler and used

widely, and thus the merit of the use of 2D-NMR in addition to 1D-NMR should be emphasized in the revised manuscript.

We thank the reviewer for this useful suggestion. As set out in the reply to comment 3 to Reviewer #1, it depends on the material if 1D or/and 2D exchange can be used to measure the exchange. The advantage of the 2D-NMR that it quantifies the exchange more unambiguously compared to the 1D exchange, and ideally both are compared to show that they yield the same results, as is shown in the present work in Figure SI8 of the supporting information.

Revisions made:

In the manuscript a short discussion is added that assesses the 1D and 2D exchange NMR methods, also in reply to comment 3 of Reviewer #1 on the versatility of the technique and the use of 1D vs 2D exchange.

2. A higher charge plateau at 2.8 V vs. Li appears in all-solid-state cell with the mixture III as shown in Fig. SI2. This plateau would be attributable to Li⁺ de-intercalation from solid electrolytes. Sulfide electrolytes such as Li₃PS₄ are reported to act as an active material by high-energy ball-milling with carbon additives [A]. The mixture III has wide contact area between electrode and electrolyte, and thus the Li₆PS₅Br electrolyte is partially used as an active material. How about the effect of de-intercalated Li₆PS₅Br on the NMR spectra?

[A] T. Hakari et al., J. Power Sources, 293 (2015) 721.

*This is an excellent suggestion by the reviewer. We have electrochemically delithiated Li₆PS₅Br, by using it as cathode (by ball milling Li₆PS₅Br with carbon additives), and performed additional NMR experiments. The resulting VT T₁ data is added in Figure SI9, showing that the T₁ appears hardly affected. This indicates that the bulk of the Li₆PS₅Br appears unaffected by the charging, oxidation of which is apparently limited to the surface, consistent with the recent findings of [Auvergniot et al. Chemistry of Materials **2017** 29 (9), 3883-3890]. See also the discussion on comment 2 of Reviewer #2*

Revisions made:

We have included reference [A] suggested by the reviewer, and added the T₁ NMR results of the charged Li₆PS₅Br to Figure SI9 in the supporting information, and a short discussion is added in the manuscript.

3. Charge-discharge profile and cycle performance of mixture IV should be added to Fig 1b and Fig. SI2.

We agree.

Revisions made:

The charge-discharge profile of mixture IV is added to supplementary Figure SI2, and the cycle performance to Figure 1b.

4. Cycle performance of all-solid-state batteries using the mixture III is shown in Fig. 1. Capacity fading is observed and this charge-discharge performance should be compared with those reported in solid-state batteries using Li₂S-electrolyte positive electrodes.

We agree.

Revisions made:

In the manuscript the charge-discharge performance is now compared to additional reports on solid-state batteries using Li_2S -electrolyte positive electrodes.

5. As shown in Table 1, a bulk resistance of the mixture III is increased by charge-discharge cycles. Why was the resistance of an electrolyte separator layer increased? A detailed explanation is needed. The bulk and interface resistances in Table 2 are different from those in Table 1. The difference is discussed in the revised manuscript.

We agree this is confusing. In Table 1 the bulk and interface resistance obtained by electrochemical impedance (EIS) is reported, whereas in Table 2 the values obtained by exchange NMR are reported. This comparison was added to illustrate that it is difficult for EIS in these all solid state batteries to quantify and distinguish bulk and interface resistance, which is the motivation to perform the present NMR experiments. The observation of the Reviewer, that the bulk resistance of mixture III appears to increase by cycling is indeed surprising, and we don't have a rational for this, other than that it illustrates the difficulty of EIS to quantify these properties in realistic all solid state batteries. The NMR T1 measurements indicate that the bulk conductivity remains constant.

Revisions made:

To avoid confusion we have moved Table 1 to the supplementary information, and attempted to make the discussion of the EIS results more clear by underlining the difference of the two tables.

6. The authors mentioned that about 20% Li-ions was moved from the $\text{Li}_6\text{PS}_5\text{Br}$ to the Li_2S in mixture III in Fig. 2I. Quantitative determination of Li-ion amount is important, but calculation process has not been described. Detailed explanation should be added in the revised manuscript.

We are very sorry that the calculation process was not described in detail.

Revision made:

A description of the calculation has been added to the manuscript: By integrating the exchange peak, and comparing this intensity to the peaks on the diagonal before exchange, the relative amount of Li-ions that has been exchange is quantified, amounting 20% in Figure II in Figure 2I.